# High-Precision Anti-Interference Control of Direct Drive Components

**Jieji Zheng** **, Xianliang Jiang, Guangan Ren, Xin Xie \* and Dapeng Fan**

College of Intelligence Science and Technology, National University of Defense Technology, Changsha 410073, China; zhengjieji@nudt.edu.cn (J.Z.); jiangxianliang@nudt.edu.cn (X.J.); 369220424@163.com (G.R.); fdp@nudt.edu.cn (D.F.)
\* Correspondence: xiexin12@nudt.edu.cn; Tel.: +86-13787043604

**Abstract:** This study presents a compound control algorithm that enhances the servo accuracy and disturbance suppression capability of direct drive components (DDCs). The servo performance of DDCs is easily affected by external disturbance and the deterioration of assembly characteristics due to a lack of deceleration device. The purpose of this study is to compensate for the impact of external and internal disturbances on the system. First, a linear state space model of the system is established. Second, we analyzed the main factors restricting the performance of DDCs which includes sensor noise, friction and external disturbance. Then, a fractional-order proportional integral (FOPI) controller was used to eliminate the steady-state error caused by the time-invariable disturbance which can also improve the system's anti-interference capability. A state-augmented Kalman filter (SAKF) was proposed to suppress the quantization noise and compensate for the time-varying disturbances simultaneously. The effectiveness of the proposed compound algorithm was demonstrated by comparative experiments, demonstrating a maximum 89.34% improvement. The experimental results show that, compared with the traditional PI controller, the FOPISAKF controller can not only improve the tracking accuracy of the system, but also enhance the disturbance suppression ability.

**Keywords:** direct drive components; disturbance suppression; fractional-order control; state-augmented Kalman filter

## 1. Introduction

Direct drive components (DDCs) are widely used in light load and high precision equipment such as photoelectric pods, seekers, small robots, etc., because of its better speed versus torque characteristics, high efficiency, high dynamic response, higher speed operating range and low maintenance cost [1,2]. Nevertheless, DDCs are susceptible to external disturbance, internal friction, and sensor noise, making it difficult to meet the equipment's stable accuracy index which reaches the micro-radian level under interference.

For the compensation of friction and external disturbance, many scholars have conducted a lot of research. The traditional PI control is the most widely used in the industrial field due to its simple structure and good robustness. However, as the system accuracy increases many nonlinear factors and strong interference during practical application occur, and traditional PI controllers have been unable to meet the requirements. On this basis, many scholars have studied the integration of various advanced intelligent control algorithms into PI control, e.g., genetic algorithm, expert system algorithm, neural network algorithm, fuzzy logic control algorithm. Although intelligent algorithms have advantages such as strong optimization capabilities, they present problems such as a high amount of calculation, long convergence time, and difficulty in implementation.

During the past decades, researchers have paid attention to fractional-order controllers [3–5]. Compared with the traditional PI controllers, the fractional-order proportional integral (FOPI) controller adds one additional design parameter, thereby providing additional degrees of freedom for the control structure. The order of integration in an FOPI controller is an adjustable parameter, which accounts for the shortcomings of PI

controllers, and it is simple to implement. Shah P reviewed the development of fractional-order controllers, including the latest research progress in the design methods, parameter tuning methods, and engineering applications [6]. His research promotes the development of FOPI controllers. For the fast dynamic response and disturbance suppression of permanent magnet synchronous motors, Jakovljević studied fractional and distributed order proportional-integral-derivative (FOPID and DOPID) controllers [7]. His research results show that FOPID and DOPID controllers have great advantages in anti-disturbance. Liu firstly proposed a combination of an FOPID controller and active disturbance rejection control (ADRC) method for current compensation and voltage tracking of active power filters [8]. The simulation results show that compared with the traditional double-loop control method, this method has greater robustness and higher compensation accuracy. Bingi proposed the two-degree-of-freedom fractional-order PID (2DOF-FOPID) controller for the real-time control of pressure process in both parallel and series configurations [9]. From the real-time experimental results obtained, the proposed approach outperforms PID, FOPID and 2DOF-PID controllers in terms of overshoot and settling time. Hence, the approach has better set-point tracking ability and disturbance rejection capability. Seyedtabaii reexamined the robustness of the fractional-order controller in containing the UAV aerodynamic parameters uncertainty and suggested modifications for better vehicle performance and lower computations [10]. He proposed a modified method for flat phase margin FOPID design and the experiment results demonstrated that the proposed method outperforms the conventionally designed FOPID and definitely PID in leading the roll, yaw and pitch motion in a coherent manner. The abovementioned research proves that the fractional-order controller has a wide range of application prospects.

Since an FOPI control introduces an adjustable integration order, finding a suitable tuning method is the focus of research to promote the application of fractional-order controllers in the industrial field. The existing tuning methods can be divided into auto-tuning, optimal tuning, and robust tuning [11–14] and frequency domain design [15–17]. To recap, the auto-tuning, optimal tuning and robust tuning are too complex and each of them is suitable only for a specific class of systems. The frequency domain design method is the easiest to implement in engineering and has better stability, so it is adopted to realize the tuning of an FOPI controller. It should be pointed out that the realization of the frequency band limitation of the fractional-order system is of equal importance in practice. XUE has conducted in-depth research in this area, and finally put forward a modified Oustaloup filter [18] to approximate the fractional integrator that is proved to be effective.

Due to the limitation of system size and quality, the DDCs are rarely equipped with a tachometer for shaft rotation speed measurement. Instead, it collects and differentiates the measured angle of an incremental encoder to obtain the angular velocity. The resolution of the incremental encoder is not high enough, causing noise in the sensor signal. In addition, DDCs have the problem of being greatly affected by external load disturbances. For these two problems, this paper proposes a method of strapping down a state-augmented Kalman filter (SAKF) [19] with the FOPI controller. Compared with the traditional Kalman filter, SAKF expands the one-dimensional state variable on the basis of the original system state space, and uses the Kalman filter to observe and compensate for external disturbances [19–21].

The contribution of this paper is mainly to propose a comprehensive control algorithm that combines a fractional-order controller and SAKF. The FOPI controller is used to eliminate linear factors in the system and improve the system's anti-interference ability, while SAKF can simultaneously realize the filtering of sensor signals with high signal-to-noise ratio and the observation of external load disturbances. The sections of this paper are organized as follows: In Section 2, a dynamic model of DDCs is established and the key factors affecting high-precision control of DDCs are analyzed. Section 3 introduces the structure of the control algorithm and expounds the design method of the FOPI controller and SAKF. The experiments are described in Section 4. Finally, the conclusion of this study is summarized in Section 5.

## 2. System Modelling and Problem Statement

### 2.1. Modelling of DDCs

The DDCs system consists of a brushless DC torque motor (BLDC), an inertial disk, an incremental encoder and a load as shown in Figure 1.

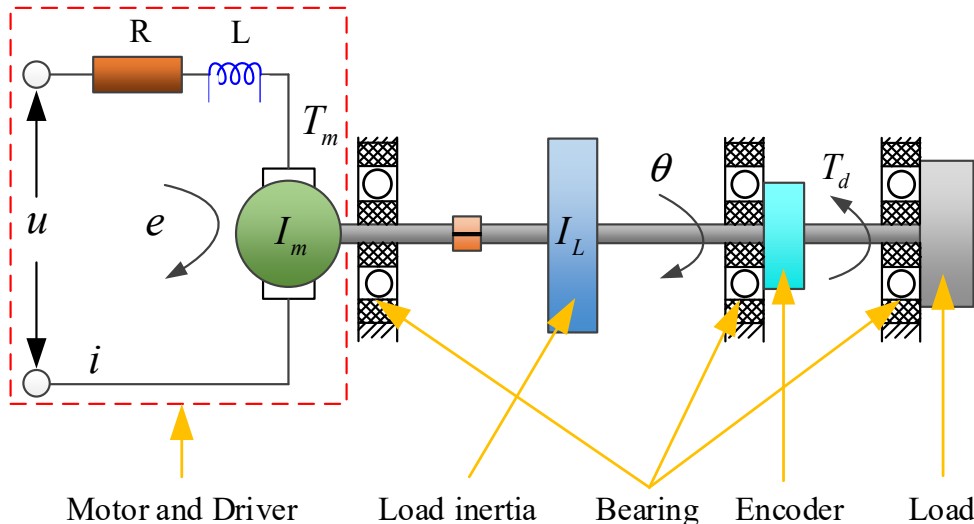

**Figure 1.** Schematic diagram of DDCs.

The symbols in Figure 1 are defined as follows: $L$ is the armature inductance of the motor, $R$ is the armature resistance, $i$ is the armature current, $u$ is the armature voltage, $e$ is the back-electromotive force (back-EMF), $I_m$ is the moment of inertia of the motor rotor, $I_L$ is the moment of inertia of the disk, $T_m$ is the motor torque, $T_d$ is the friction disturbance torque and $\theta$ is the angular of the shaft.

The dynamical equations of the system was obtained as follows:

$$\begin{cases} L\frac{di}{dt} + Ri = u - e, \\ e = K_e\dot{\theta}, \\ I\ddot{\theta} + B\dot{\theta} = T_m - T_d, \\ T_m = K_m i \end{cases} \tag{1}$$

where $I = I_m + I_L$ is the overall moment of inertia of the system, $K_e$ is back-EMF coefficient, $K_m$ is the motor torque constant and $B$ is the damping coefficient of the system.

Generally, the driver is set in the current mode and the bandwidth of the current loop is much higher than the bandwidth of the speed loop (more than ten times). Therefore, the current loop can be viewed as a proportional element, i.e., $i = K_D u$, where $K_D$ is the conversion factor of the driver. Hence, the DDCs model can be simplified as shown in Figure 2.

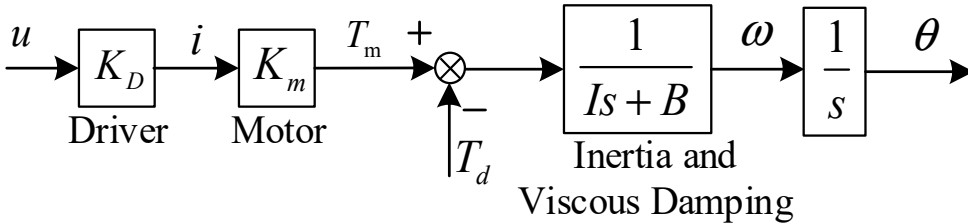

**Figure 2.** Simplified linear model of DDCs, where $\omega$ is the angular velocity of the shaft.

According to Figure 2, the open-loop transfer function of the DDCs can be expressed as:

$$\omega(s) = \frac{1}{Is + B}(K_m K_D u - T_d)$$ (2)

In order to simplify the observation of torque disturbance, the disturbance is converted to the command end. Thus we have $\zeta(s) = T_d(s)/(K_m K_D)$. We defined $\alpha = K_m K_D / I$ and $\beta = -B/I$. Therefore, the relationship between the angular velocity $\omega$, the control command $u$ and the equivalent input disturbance $\zeta$ can be converted into:

$$\omega(s) = \frac{\alpha}{s - \beta}(u(s) - \zeta(s))$$ (3)

The angular $\theta$ of the shaft can be expressed as:

$$\theta(s) = \frac{1}{s}\omega(s) = \frac{1}{s}\frac{\alpha}{s - \beta}(u(s) - \zeta(s))$$ (4)

After converting the above transfer function into state space, we obtain:

$$\begin{bmatrix} \dot{\theta}(t) \\ \dot{\omega}(t) \end{bmatrix} = A_c \begin{bmatrix} \theta(t) \\ \omega(t) \end{bmatrix} + \begin{bmatrix} B_c & -B_c \end{bmatrix} \begin{bmatrix} u(t) \\ \zeta(t) \end{bmatrix} \text{ where } A_c = \begin{bmatrix} 0 & 1 \\ 0 & \beta \end{bmatrix}, B_c = \begin{bmatrix} 0 \\ \alpha \end{bmatrix}$$ (5)

Since the control signal $u$ is generated by a D/A converter, Equation (3) can be rewritten in the discrete-time domain with a zero order hold at the input stage as [20,22]

$$\omega(k) = \frac{\alpha_d}{z - \beta_d}(u(k) - \zeta(k)), \text{ where } \alpha_d = \frac{\alpha}{-\beta}(1 - e^{\beta t_s}), \beta_d = e^{\beta t_s}$$ (6)

where $k$ is the sample counter, $\alpha_d$ is the discrete time transfer function gain, $\beta_d$ is the discrete time pole, $z$ is the forward shift operator and $t_s$ is the sampling period. Then, the discrete time version of the state space expression in Equation (5) can be obtained as:

$$\begin{bmatrix} \theta(k+1) \\ \omega(k+1) \end{bmatrix} = A_d \begin{bmatrix} \theta(k) \\ \omega(k) \end{bmatrix} + \begin{bmatrix} B_d & -B_d \end{bmatrix} \begin{bmatrix} u(k) \\ \zeta(k) \end{bmatrix}, \text{ where } A_d = e^{A_c t_s}, B_d = \int_0^{t_s} e^{A_c \lambda}d\lambda \cdot B_c$$ (7)

Restricted by the level of sensor manufacturing accuracy, the sensor inevitably introduces quantization noise and measurement errors, which would affect the accuracy and stability of servo control. The noise is mainly composed of D/A converter quantization noise and angular measurement quantization noise.

The real output voltage of the D/A converter can be expressed as:

$$u = u' + \widetilde{u}$$ (8)

where $u'$ is the voltage command and $\widetilde{u}$ is the voltage noise. The distribution and magnitude of the noise depends on the accuracy of the D/A converter.

Assuming that the noise distribution of the output voltage is approximately a uniform probability distribution, the variance of the D/A converter noise can be obtained as:

$$R_{\widetilde{u}} = E\left\{(\widetilde{u} - E[\widetilde{u}])^2\right\} = \frac{(\delta u)^2}{12}$$ (9)

where $\delta u$ is the resolution of D/A converter.

Similarly, the angular $\theta_m$ and the angular velocity $\omega_m$ measured by the encoder are expressed, respectively:

$$\theta_m(k) = \theta(k) + \widetilde{\theta}(k), \omega_m(k) = \omega(k) + \widetilde{\omega}(k)$$ (10)

where $\theta$ is the real output angular, $\widetilde{\theta}$ is the angular noise, $\omega$ is the real output angular velocity, $\widetilde{\omega}$ is the angular velocity noise. The variances of the angular noise and angular velocity noise are:

$$R_{\widetilde{\theta}} = E\left\{(\widetilde{\theta} - E[\widetilde{\theta}])^2\right\} = \frac{(\delta\theta)^2}{12}, R_{\widetilde{w}} = E\left\{(\widetilde{\omega} - E[\widetilde{\omega}])^2\right\} = \frac{(\delta\omega)^2}{12} \tag{11}$$

where $\delta\theta$ is the resolution of the encoder. Since the angular velocity $\omega$ is derived from the angular $\theta$, the relationship between $\delta\theta$ and $\delta\omega$ can be expressed as $\delta\omega = \delta\theta/t_s$.

In order to design an unbiased state estimator, the influence of disturbance characteristics on the system must also be considered in the dynamic model. Friction is the dominant disturbance of the shaft control system, and it does not change suddenly during steady-state motion. Therefore, it can be reasonably assumed that the disturbance $\zeta(k)$ is composed of a piecewise constant signal with zero average white noise perturbation $\zeta_d(k)$ as below:

$$\zeta(k+1) = \zeta(k) + \zeta_d(k) \tag{12}$$

where the variance of the disturbance perturbation $R_{\zeta d}$ is the tuning parameter of the Kalman Filter. Combining Equations (6)–(11), an augmented discrete-time state-space model including linear dynamics, input and measurement noise and disturbance models is deduced:

$$\begin{cases} \begin{bmatrix} \theta(k+1) \\ \omega(k+1) \\ \zeta(k+1) \end{bmatrix} = A_{aug} \begin{bmatrix} \theta(k) \\ \omega(k) \\ \zeta(k) \end{bmatrix} + B_{aug} \begin{bmatrix} u(k) \end{bmatrix} + W_{aug} \begin{bmatrix} \widetilde{u}(k) \\ \zeta_d(k) \end{bmatrix} \\ \begin{bmatrix} \theta_m(k) \\ \omega_m(k) \end{bmatrix} = C_{aug} \begin{bmatrix} \theta(k) \\ \omega(k) \\ \zeta(k) \end{bmatrix} + V_{aug} \begin{bmatrix} \widetilde{\theta}(k) \\ \widetilde{\omega}(k) \end{bmatrix} \end{cases} \tag{13}$$

where,

$$A_{aug} = \begin{bmatrix} A_d & -B_d \\ 0 \quad 0 & 1 \end{bmatrix}, B_{aug} = \begin{bmatrix} B_d \\ 0 \end{bmatrix}, C_{aug} = \begin{bmatrix} 1 & 0 & 0 \\ 0 & 1 & 0 \end{bmatrix}, W_{aug} = \begin{bmatrix} B_d & 0 \\ & 0 \\ 0 & 1 \end{bmatrix}, V_{aug} = \begin{bmatrix} 1 & 0 \\ 0 & 1 \end{bmatrix}$$

In the above model, $A_{aug}$ and $B_{aug}$ are the augmented system and input matrices, respectively. The disturbance $\zeta(k)$ appears as a state which is estimated by the Kalman filter. $W_{aug}$ decides how the process noise vector $\begin{bmatrix} \widetilde{u} & \zeta_d \end{bmatrix}^T$ affects the state transition. $C_{aug}$ is the output matrix, and $V_{aug}$ relates the measurement noise $\begin{bmatrix} \widetilde{\theta} & \widetilde{\omega} \end{bmatrix}^T$ to the measured angular and angular velocity $\begin{bmatrix} \theta_m & \omega_m \end{bmatrix}^T$, which is also referred to as the output vector. The above model is used in the SAKF design for disturbance compensation in Section 3.2.

### 2.2. Problem Statement

From the above analysis, it can be seen that the disturbance contained in the DDCs mainly include into two parts. The first part is the quantization noise $R_{\widetilde{u}}, R_{\widetilde{\theta}}, R_{\widetilde{w}}$, which is introduced by the D/A converter and the measuring device used in the system. The other is the uncertain disturbance $T_d$ composed of the friction torque, the mass unbalance torque and external load disturbance. For brevity, we did not analyze these disturbances in detail. Additionally, the uncertain disturbance can be divided into time-invariable disturbance $\overline{T}_d$ and time-varying disturbance $\widetilde{T}_d$ [23,24].

Time-invariable disturbance $\overline{T}_d$ and time-varying disturbance $\widetilde{T}_d$ will introduce the steady-state error and dynamic error, respectively in the control process. Excessive measurement noise may cause an instability of the control system. Therefore, the above three factors

must be managed accordingly to improve the control accuracy and the anti-interference capability of the system.

For $\overline{T}_d$, which appears as a step signal, the FOPI controller is useful to eliminate the influence of its effect as well as improve the ability to resist external disturbances.

For $\widetilde{T}_d$, the FOPI controller is not feasible because the frequency component is unknown. A state augmented Kalman filter (SAKF) has been proven to be an effective method to realize observation and compensation of $\widetilde{T}_d$ and the filtering of measurement noises.

## 3. Controller Design and Analysis

In this section, a compound control strategy is proposed to improve the nonlinear dynamic behavior due to time-invariable disturbance $\overline{T}_d$, time-varying disturbance $\widetilde{T}_d$ and measurement noise.

As shown in Figure 3, the fractional-order controller is used to eliminate steady-state errors caused by $\overline{T}_d$. For $\widetilde{T}_d$ and measurement noise, an SAKF observer is configured to track the residual perturbation by collecting the input voltage $u$, feedback angular velocity $\omega_m$ and feedback angular $\theta_m$. Then, the estimated disturbance $\hat{T}_d$ after a coefficient $K_g = \frac{1}{K_D K_m}$ is added to the input voltage through feedforward and the filtered angular velocity $\hat{\omega}$ is used for speed feedback control.

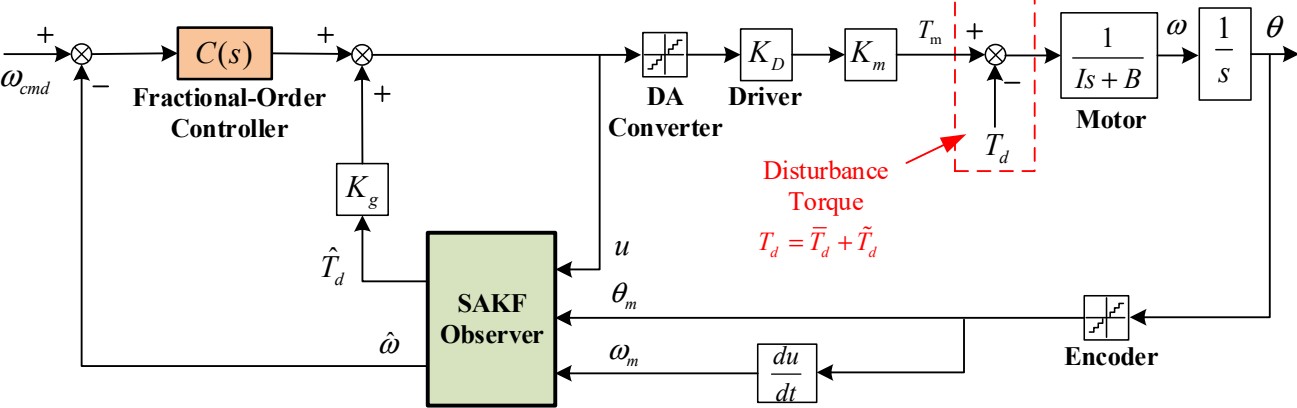

**Figure 3.** Block diagram of the proposed Compound control strategy.

### 3.1. Design of FOPI Controller

As shown in Figure 2, the open-loop linear system model of velocity can be expressed as:

$$G_{open}(s) = \frac{K_m K_D}{Is + B} \tag{14}$$

Let $s = jw$, the expression of the transfer function in frequency domain is:

$$G_{open}(jw) = \frac{K_m K_D}{jIw + B} = \frac{K_m K_D (B - jIw)}{I^2 w^2 + B^2} \tag{15}$$

The phase and amplitude of the transfer function are:

$$\text{Arg}\left[G_{open}(jw)\right] = -\arctan\left(\frac{Iw}{B}\right) \tag{16}$$

$$\left|G_{open}(jw)\right| = \frac{K_m K_D}{I^2 w^2 + B^2} \sqrt{I^2 w^2 + B^2} = \frac{K_m K_D}{\sqrt{I^2 w^2 + B^2}} \tag{17}$$

Set the fractional-order controller as a FOPI controller which is expressed as:

$$C(s) = FOK_P\left(1 + FOK_i \frac{1}{s^\lambda}\right) \tag{18}$$

where $FOK_P$, $FOK_i$ and $\lambda$ are the proportional coefficient, integral coefficient and order of the integrator of the FOPI controller, respectively.

Similarly, let $s = jw$, we obtain:

$$
\begin{aligned}
C(jw) &= FOK_P(1 + FOK_i \frac{1}{(jw)^\lambda}) \\
&= FOK_P[(1 + FOK_i \cdot w^{-\lambda} \cos(-\tfrac{\lambda\pi}{2})) - j(FOK_i \cdot w^{-\lambda} \sin(-\tfrac{\lambda\pi}{2}))]
\end{aligned}
\tag{19}
$$

The phase and amplitude of the FOPI controller are:

$$
\text{Arg}[C(jw)] = -\arctan\left( \frac{FOK_i \cdot w^{-\lambda} \sin\left(-\tfrac{\lambda\pi}{2}\right)}{1 + FOK_i \cdot w^{-\lambda} \cos\left(-\tfrac{\lambda\pi}{2}\right)} \right)
\tag{20}
$$

$$
|C(jw)| = FOK_P \cdot \sqrt{\left(1 + FOK_i \cdot w^{-\lambda} \cos\left(-\frac{\lambda\pi}{2}\right)\right)^2 + \left(FOK_i \cdot w^{-\lambda} \sin\left(-\frac{\lambda\pi}{2}\right)\right)^2}
\tag{21}
$$

The velocity loop open-loop transfer function of the system in frequency domain is $G(jw) = C(jw) \cdot G_{open}(jw)$. Combining Equations (15) and (19), the phase of $G(jw)$ can be obtained as:

$$
\text{Arg}[G(jw)] = -\arctan\left(\frac{Iw}{B}\right) - \arctan\left( \frac{FOK_i \cdot w^{-\lambda} \sin\left(-\tfrac{\lambda\pi}{2}\right)}{1 + FOK_i \cdot w^{-\lambda} \cos\left(-\tfrac{\lambda\pi}{2}\right)} \right)
\tag{22}
$$

In the same way, combining Equations (16) and (20), the amplitude of $G(jw)$ is shown as:

$$
|G(jw)| = \frac{K_m K_D \cdot FOK_P}{\sqrt{I^2 w^2 + B^2}} \sqrt{\left(1 + FOK_i \cdot w^{-\lambda} \cos\left(-\frac{\lambda\pi}{2}\right)\right)^2 + \left(FOK_i \cdot w^{-\lambda} \sin\left(-\frac{\lambda\pi}{2}\right)\right)^2}
\tag{23}
$$

In order to obtain the three parameters of the FOPI controller, three constraint equations need to be established. We adopt the three criteria proposed by Monje C A [12,16] to obtain the three parameters.

A.　Phase margin specification:

$$
\text{Arg}[G(jw_c)] = -\arctan\left(\frac{Iw_c}{B}\right) - \arctan\left( \frac{FOK_i \cdot w_c^{-\lambda} \sin\left(-\tfrac{\lambda\pi}{2}\right)}{1 + FOK_i \cdot w_c^{-\lambda} \cos\left(-\tfrac{\lambda\pi}{2}\right)} \right) = -\pi + \varphi_m
\tag{24}
$$

where $w_c$ is the gain crossover frequency and $\varphi_m$ is the phase margin required.

B.　Robustness to variation in the gain of the plant:

$$
\left. \frac{d(\text{Arg}[G(jw)])}{dw} \right|_{w=w_c} = 0
\tag{25}
$$

That is, the derivative of $G(jw)$ with respect to the frequency is forced to be zero at the gain crossover frequency so that the closed-loop system is robust in gaining variations, and therefore the overshoots of the response are almost invariant.

C.　Gain crossover frequency specification:

$$
|G(jw_c)| = \frac{K_m K_D \cdot FOK_P}{\sqrt{I^2 w_c^2 + B^2}} \sqrt{\left(1 + FOK_i \cdot w_c^{-\lambda} \cos\left(-\frac{\lambda\pi}{2}\right)\right)^2 + \left(FOK_i \cdot w_c^{-\lambda} \sin\left(-\frac{\lambda\pi}{2}\right)\right)^2} = 1
\tag{26}
$$

Combining Equations (23) and (24), the coefficient $FOK_i$ and order $\lambda$ of the integrator can be obtained as Equations (26) and (27):

$$FOK_i = \frac{\tan\left(\pi - \varphi_m - \arctan\left(\frac{Iw_c}{B}\right)\right)}{w_c^{-\lambda}\sin\left(\frac{\lambda\pi}{2}\right) - w_c^{-\lambda}\cos\left(\frac{\lambda\pi}{2}\right)\tan\left(\pi - \varphi_m - \arctan\left(\frac{Iw_c}{B}\right)\right)} \quad (27)$$

$$FOK_i = \frac{-N \pm \sqrt{N^2 - 4Mw_c^{-2\lambda}}}{2Mw_c^{-2\lambda}} \quad (28)$$

where,

$$M = \frac{\frac{I}{B}}{1 + \left(\frac{Iw_c}{B}\right)^2}, N = 2Mw_c^{-\lambda}\cos(\frac{\lambda\pi}{2}) - \lambda w_c^{-\lambda}\sin(\frac{\lambda\pi}{2})$$

According to Equations (26) and (27), when the gain crossover frequency $w_c$ and the phase margin $\varphi_m$ are given, we take the order of the integrator $\lambda$ as the independent variable and the coefficient of the integrator $FOK_i$ as the dependent variable. Two curves can be drawn and the intersection of the curves is the tuning result.

According to Equation (25), the proportional coefficient of the fractional controller $FOK_p$ can be expressed as:

$$FOK_p = \frac{\sqrt{I^2w_c^2 + B^2}}{K_m K_D \sqrt{\left(1 + FOK_i \cdot w_c^{-\lambda}\cos\left(-\frac{\lambda\pi}{2}\right)\right)^2 + \left(FOK_i \cdot w_c^{-\lambda}\sin\left(-\frac{\lambda\pi}{2}\right)\right)^2}} \quad (29)$$

Therefore, as long as the gain crossover frequency $w_c$ and the phase margin $\varphi_m$ are known, the parameter values of the FOPI controller can be calculated according to the above three criteria.

Since the FOPI controller is an infinite-dimensional system, it needs to memorize all the historical inputs in the past, which is difficult in practical engineering applications. The band limit implementation of fractional-order systems is important in practice, which is why Xue [18] proposed a modified Oustaloup filter to approximate the fractional integrator:

$$s^\lambda \approx K_0 \left(\frac{ds^2 + hw_h s}{d(1-\lambda)s^2 + hw_h s + d\lambda}\right) \prod_{k=-n}^{n} \frac{s + w'_k}{s + w_k} \quad (30)$$

where,

$$K_0 = \left(\frac{dw_b}{h}\right)^\lambda \prod_{k=1}^{n} \frac{w_k}{w'_k} \quad (31)$$

where, $k = 1, 2, \ldots, n$, $w'_k = w_b w_u^{(2k-1-\lambda)/n}$, $w_k = w_b w_u^{(2k-1+\lambda)/n}$, $w_u = \sqrt{\frac{w_h}{w_b}}$.

where the frequency range to be fit is defined as $(w_b, w_h)$, $n$ is the order of the filter.

### 3.2. Design of SAKF Based Estimator

The time-varying disturbance $\widetilde{T}_d$ and measurement noise of DDCs are the main factors affecting servo performance. To improve the disturbance suppression capability of DDCs, a compound control strategy that strapping the SAKF observer behind the FOPI controller is proposed. The SAKF can realize the observation of nonlinear disturbances and enhance the signal-to-noise ratio of the sensor signal at the same time.

Equation (12) establishes a discrete extended state space including a dynamic model and a noise model. Furthermore, the form of SAKF is obtained as [19,21]:

$$\begin{bmatrix} \hat{\theta}(k) \\ \hat{\omega}(k) \\ \hat{\zeta}(k) \end{bmatrix} = (E - K_{obs}C)A_{aug}\begin{bmatrix} \hat{\theta}(k-1) \\ \hat{\omega}(k-1) \\ \hat{\zeta}(k-1) \end{bmatrix} + (E - K_{obs}C_{aug})B_{aug}[u(k-1)] + K_{obs}\begin{bmatrix} \theta_m(k) \\ \omega_m(k) \end{bmatrix} \qquad (32)$$

where $E$ is the identity matrix. The gain $K_{obs}$ of the SAKF can be obtained offline iteratively by Equation (32):

$$H(k|k-1) = A_{aug}H(k-1|k-1)A_{aug}^T + W_{aug}R_zW_{aug}^T$$
$$K_{obs}(k) = H(k|k-1)C^T\left[R_v + C_{aug}H(k|k-1)C_{aug}^T\right]^{-1} \qquad (33)$$
$$H(k|k) = \left[E - K_{obs}(k)C_{aug}\right]H(k|k-1)$$

where $H$ is the covariance of state estimation error and $R_z = \begin{bmatrix} R_{\tilde{u}} & 0 \\ 0 & R_{\zeta d} \end{bmatrix}, R_v = \begin{bmatrix} R_{\tilde{\theta}} & 0 \\ 0 & R_{\tilde{\omega}} \end{bmatrix}$.

## 4. Experiment

In order to demonstrate the effectiveness of the proposed compound control strategy, comparative experiments between the traditional methods and the proposed one were carried out.

### 4.1. Experiment Setup

The experimental device is mainly composed of a BLDC, a servo driver, an adjustable inertia disk, an encoder, a magnetic powder brake, a dSPACE1103 and computer as shown in Figure 4. The inertia plate is used to simulate load inertia and the external disturbance is applied by the magnetic powder brake. The rotational speed of the shaft is measured by an incremental encoder. All the required parameters are provided in Table 1.

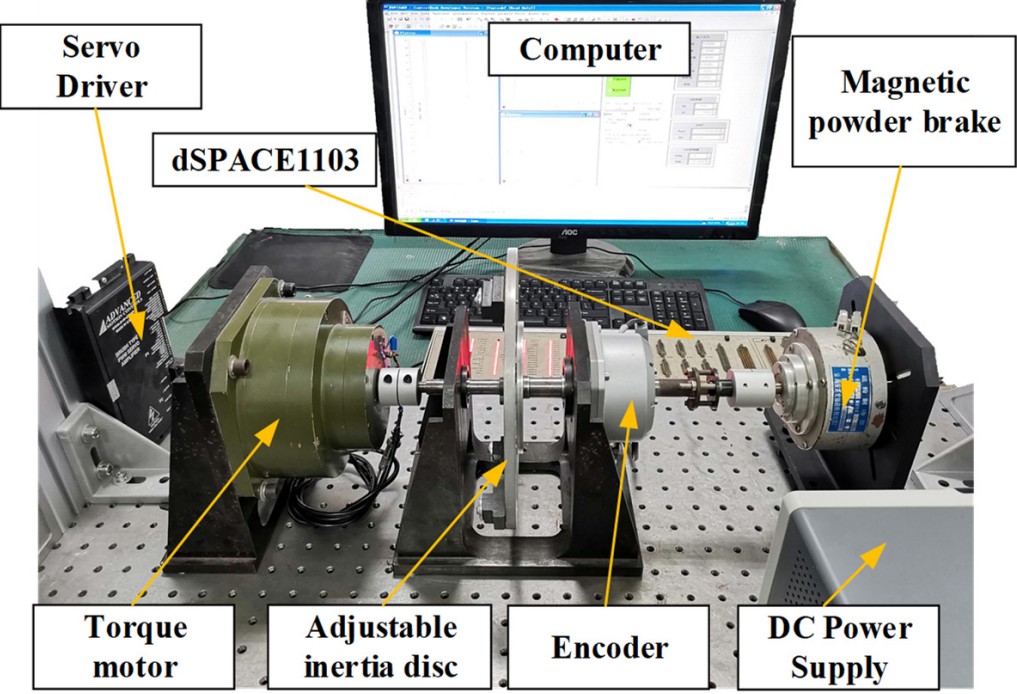

**Figure 4.** Experimental setup.

**Table 1.** Parameters of the experimental device.

| Symbol | Description | Value |
|---|---|---|
| $I_m/(\text{kg}\cdot\text{m}^2)$ | Moment of inertia of motor rotor | $6.5 \times 10^{-3}$ |
| $I_L/(\text{kg}\cdot\text{m}^2)$ | Moment of inertia of load | $2.3 \times 10^{-3}$ |
| $B/(\text{N}\cdot\text{m}\cdot\text{s}\cdot\text{rad}^{-1})$ | Damping coefficient of motor rotor | 0.044 |
| $K_m/(\text{N}\cdot\text{m}\cdot\text{A}^{-1})$ | Motor torque constant | 0.73 |
| $K_D/(\text{A}\cdot\text{V}^{-1})$ | Driver conversion factor | 0.47 |
| $\delta\widetilde{u}/(\text{V})$ | DA conversion resolution | $20/(2^{16})$ |
| $\delta\widetilde{\theta}/(°)$ | Encoder resolution | 0.02 |

*4.2. Control Parameters*

According to the mechanical parameters and sensor parameters, the three parameters of the FOPI controller are first calculated.

On the basis of previous engineering experience, when the open-loop gain crossover frequency and phase margin are $w_c = 90$ rad, $\varphi_m = 45°$, the dynamic performance requirements of the system can be met. By substituting the two parameters into Equations (26) and (27), the two curves in Figure 5 were derived.

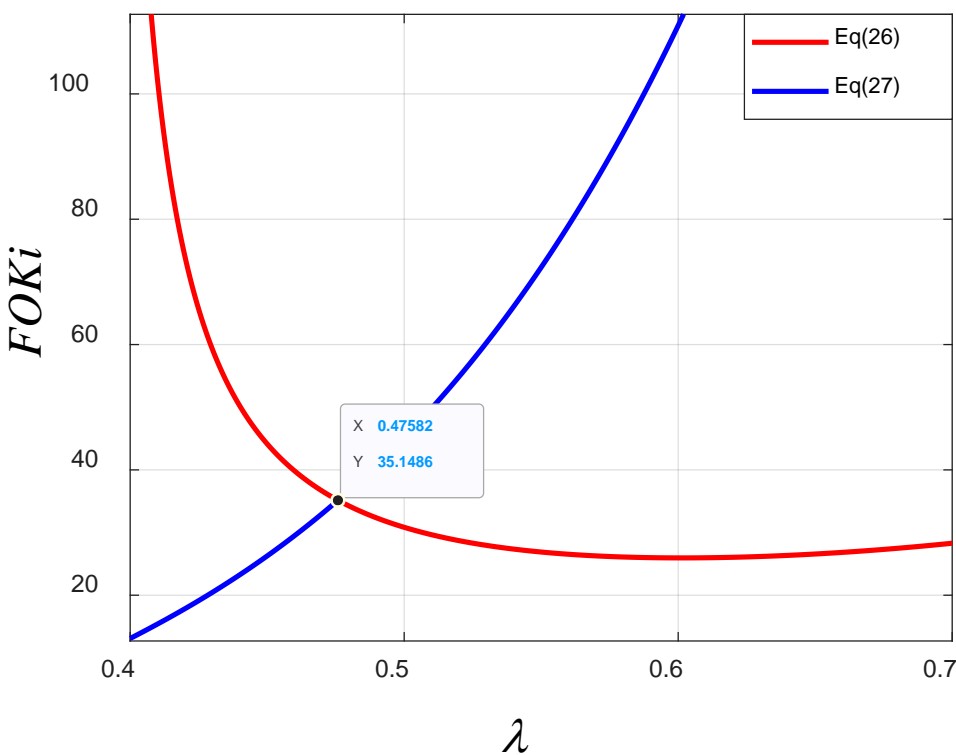

**Figure 5.** The tuning result of $FOK_i$ and $\lambda$.

From Figure 5, we find $\lambda = 0.47582$ and $FOK_i = 35.1486$. Substituting $\lambda$ and $FOK_i$ into Equation (28), $FOK_p = 0.4707$ is obtained. Therefore, the FOPI controller can be expressed as:

$$C(s) = 0.4707 \cdot \left(1 + 35.1486 \cdot \frac{1}{s^{0.47582}}\right)$$

Secondly, the approximate calculation of the integrator of the FOPI controller is carried out. In order to weigh the degree of approximation and computational efficiency of the modified Oustaloup filter to the fractional integrator, the filter order is set to $N = 9$. Refer to reference [18], the frequency range $(w_b, w_h)$ of the filter is set to (0.01, 1000). S = By substituting the above values into Equations (29) and (30), the parameter values of the modified Oustaloup filter are obtained.

Finally, the gain of SAKF $K_{obs}$ is calculated offline. After substituting the parameters in Table 1 into Equation (12), the discrete state space matrix containing the dynamic model and the noise model is obtained as:

$$A_{aug} = \begin{bmatrix} 1 & 9.975 \times 10^{-4} & -0.0011 \\ 0 & 0.995 & -2.2283 \\ 0 & 0 & 1 \end{bmatrix}, B_{aug} = \begin{bmatrix} 0.0011 \\ 2.2283 \\ 0 \end{bmatrix}, C_{aug} = \begin{bmatrix} 1 & 0 & 0 \\ 0 & 1 & 0 \end{bmatrix}, W_{aug} = \begin{bmatrix} 0.0011 & 0 \\ 2.2283 & 0 \\ 0 & 1 \end{bmatrix}, V_{aug} = \begin{bmatrix} 1 & 0 \\ 0 & 1 \end{bmatrix}$$

In the debugging, for weighing the fast response of the observer and the noise, the value of the noise and disturbance perturbation covariance is set as $R_{\zeta d} = 0.01$ and the coefficient $K_g = 2.9146$. After the offline iteration shown in Equation (32), the final optimal gains of SAKF is obtained:

$$K_{obs} = \begin{bmatrix} 0.1604 & 1.2511 \times 10^{-5} \\ 12.5111 & 0.0013 \\ -0.2022 & -2.4812 \times 10^{-5} \end{bmatrix}$$

*4.3. Experimental Results*

To verify the advancement of the proposed compound control algorithm, performance comparison experiments under different control methods were conducted. The control methods include: (1) PI, (2) FOPI and (3) FOPI + SAKF (FOPISAKF). To ensure the validity of the comparison, we used the method described in Section 3.1 to obtain the parameters of the PI controller.

Firstly, we compared the tracking performance of the four methods on the sinusoidal signal with different frequency ($20°/s$ and 1 Hz and $20°/s$ and 5 Hz) and the results are shown in Figure 6a,b. Secondly, we tested the step response characteristics of the system under different control methods whose results are given in Figure 6c. Thirdly, Figure 6d compares the anti-interference performance of the three control methods.

In Figure 6, the green curve represents the result of PI control, the dark blue curve represents the result of FOPI control and the pink represents the result of FOPISAKF control.

The root-mean-square-err (RMSE) is introduced to quantify the tracking error of the three experiments in Figure 6 and the performance comparison results are shown in Table 2.

From the above experimental results, it can be seen that, compared with the traditional PI controller, FOPISAKF controllers achieve an improvement of 80.58%, 66.41%, 46.62% and 89.34%, respectively. Hence, the FOPISAKF controller can not only improve the tracking accuracy of the system, but also enhance the disturbance suppression ability.

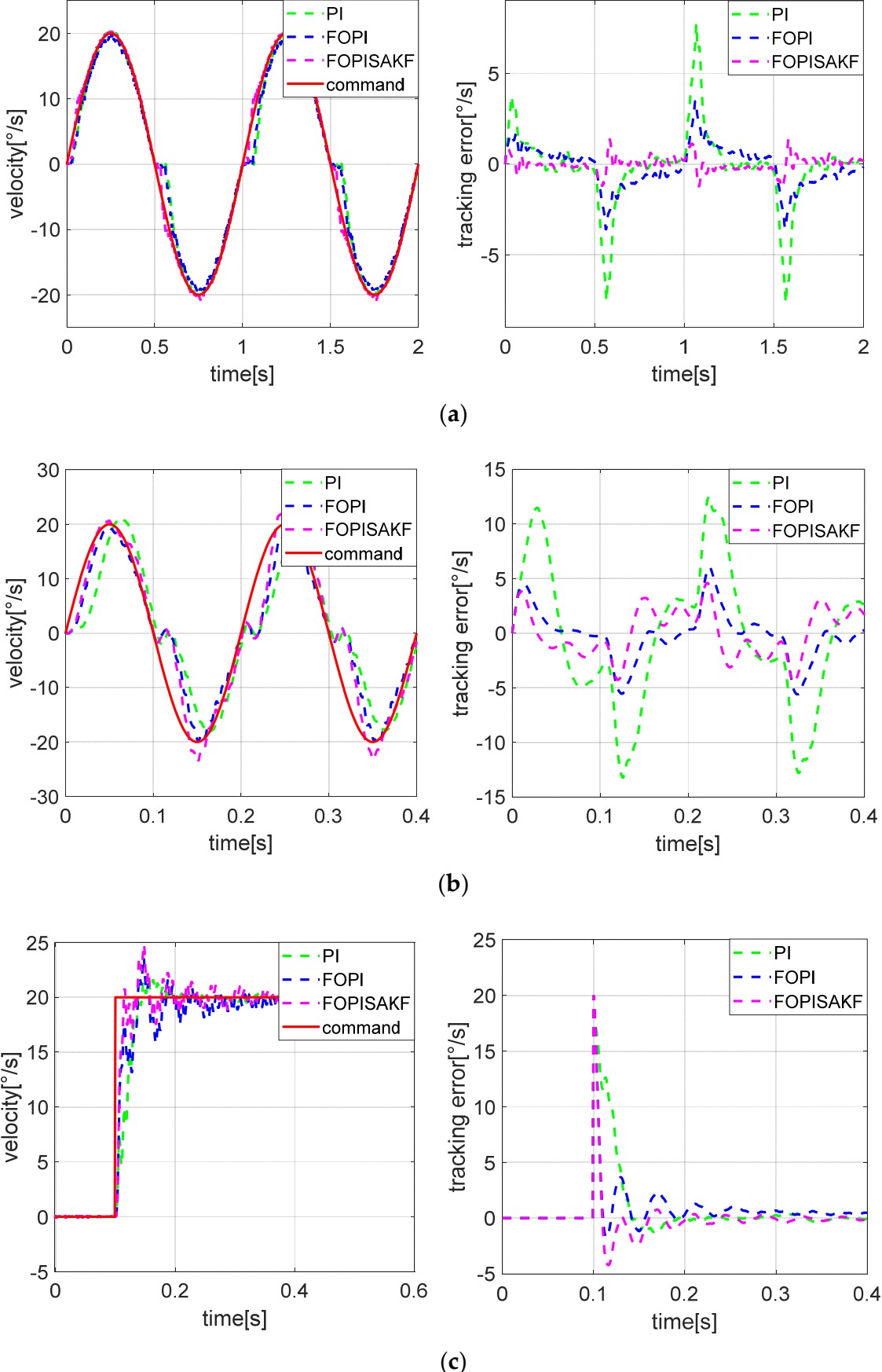

**Figure 6.** *Cont.*

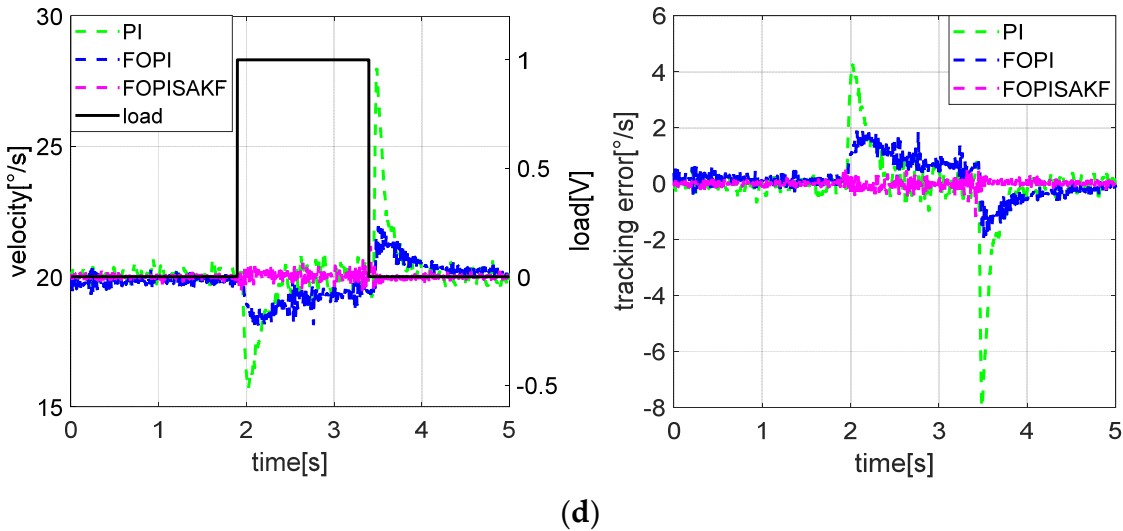

**(d)**

**Figure 6.** Performance comparison under different methods. (**a**) 20°/s&1Hz sinusoidal signal tracking ex-periment. (**b**) 20°/s&5Hz sinusoidal signal tracking experiment. (**c**) Step response experiment. (**d**) Anti-interference experiment.

**Table 2.** The RSME of experiments under different methods.

| Methods<br>Experiments | PI (°/s) | FOPI (°/s) | FOPISAKF (°/s) | Improvement (%) |
|---|---|---|---|---|
| 20°/s and 1 Hz sinusoidal signal tracking experiment | 2.06 | 1.12 | 0.40 | 80.58 |
| 20°/s and 5 Hz sinusoidal signal tracking experiment | 6.46 | 2.31 | 2.17 | 66.41 |
| Step response experiment | 2.66 | 1.69 | 1.42 | 46.62 |
| Anti-interference experiment | 1.22 | 0.63 | 0.13 | 89.34 |

## 5. Conclusions

In order to improve the speed tracking accuracy and disturbance suppression capability of DDCs, this paper proposes a compound control algorithm of an FOPI controller strapdowning with an SAKF observer. A dynamic model considering disturbance and sensor noise was established. The disturbance in the system was divided into time-invariable and time-varying parts. The FOPI controller was proposed to eliminate the time-invariable disturbance of the system and improve the anti- interference ability of the system. Furthermore, the encoder noise and external disturbance were filtered and compensated for simultaneously by the SAKF observer. The experimental results show that, compared with the traditional PI controller, the best improvement is 89.34%. The compound control algorithm can effectively improve the system servo accuracy and leads to better disturbance observation and suppression effects.

**Author Contributions:** Conceptualization, J.Z., G.R. and X.X.; Data curation, J.Z.; Funding acquisition, D.F.; Methodology, J.Z., X.J. and D.F.; Supervision, X.J.; Validation, J.Z.; Writing—original draft, J.Z.; Writing—review and editing, D.F. All authors have read and agreed to the published version of the manuscript.

**Funding:** This work was funded by National Key R&D Program of China (Grant No. 2019YFB2004700) and the National Natural Science Foundation of China (Grant No. U19A2072).

**Institutional Review Board Statement:** Not applicable.

**Informed Consent Statement:** Not applicable.

**Data Availability Statement:** Not applicable.

**Conflicts of Interest:** The authors declare no conflict of interest.

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
