# Peer review of "High-Precision Anti-Interference Control of Direct Drive Components"

_actuators, doi:10.3390/act11030095_

Round 1

Reviewer 1 Report

The paper presents aspects regarding the achievement of positioning accuracy for direct drive systems. The paper seeks to bring a degree of novelty in the field of position control and electric actuation systems by combining the use of the FOPI controller and the state augmented Kalman filter (SAKF).

Given that control over the rotor position accuracy of electric motors is a highly debated topic in the literature, the paper comes with a combination of elements on control algorithms to raise the novelty of the scientific research. If most control algorithms are based on the PID controller this work uses a more complex fractional-order integral proportional controller. Optimizing parameters for (FOPID) is very important. This type of controller has several parameters compared to the PID structure. Consequently, it has a high complexity than PID and can get a relatively better response.

However, using FOPI also the difficulty of optimizing FOPID parameters increases. A comparative study was also conducted to highlight the advantage of using a FOPI controller over the PI control scheme for adjusting the accuracy of direct drive systems in a DC motor. Extensive simulation results indicate that the controller has different characteristics when optimizing parameters using different performance criteria. And FOPI has better dynamic performance, stronger robustness than PID.

In sections 2 and 3 the mathematical model of the whole system is presented. Even if the mathematical model of the current motor is quite well known in the specialized literature, the realization of the DC model in combination with the SAKF filter and the FOPI controller is difficult to realize and presents a high degree of complexity. The FOPI control has a high degree of complexity and yet it is presented in the paper in combination with SAKF filter.

The conclusion section presents the results after implementing the presented method on the experimental model which led to much improved results, however in the future it would be interesting to see the whole model using system identification.

Reviewer 2 Report

This paper presents a compound control algorithm using FOPI and SAKF method to enhances the servo accuracy and disturbance rejection capability of DDC system. Experimental results are given to demonstrate the effectiveness of the proposed method.

Some comments and suggestions are given as follow:

  1. The language of this paper should be further polished. E.g. in line 22, "which show a maximum 89.34% improvement". should be "which shows".
  2. In the Introduction, the motivation and novelties are not clear, which should be further highlighted in the revision. In detail, the system is linear, of low dimensions in both the number of states and number of inputs and outputs. I wonder why a simple PID control is not enough for the proposed goals? One reason that springs to mind is the lack of robustness. The authors should comment on this issue to make the manuscript more self-contained.
  3. What are the advantages of the SAKF method over traditional Kalman filter method. Some theoretical analysis and experiments are suggested.
  4. Please pay attention to the layout of the article. There is a large blank on page 12.

Round 2

Reviewer 1 Report

The manuscript has been sufficiently improved and can be publish .

Reviewer 2 Report

The authors have answered my questions properly, I agree to accept its publication in current form.